# OpenReview forum: "AutoAdvExBench: Benchmarking Autonomous Exploitation of Adversarial Example Defenses"
_ICML.cc/2025/Conference — ICML 2025 oral_

### Official Review · Reviewer_Y1yh · 2025-03-12

**Overall Recommendation:** 4

**Summary:**

This paper introduced a benchmark for evaluating LLMs' ability to autonomously bypass adversarial defenses. Unlike existing benchmarks, which predominantly comprise small-scale proxy tasks, this framework provides a more rigorous assessment of LLMs' capacity to replicate tasks typically conducted by security professionals. Additionally, the authors proposed a novel agent-based methodology for circumventing real-world adversarial defenses within the proposed framework.

**Claims And Evidence:**

The study makes the following key claims:
(1) a large-scale benchmark dataset (redacted for review);
(2) a strong LLM agent designed for security analysis; and
(3) a benchmark framework that systematically evaluates LLM agents’ capabilities as security experts.


These claims are substantiated through:
(1) the authors’ commitment to releasing the benchmark post-review
(2) comprehensive empirical results demonstrating the agent’s performance on the benchmark, as detailed in the paper; and
(3) a rigorously designed methodology that validates the benchmark’s usefulness

**Essential References Not Discussed:**

No.

**Experimental Designs Or Analyses:**

The experimental design and analytical methods employed in this study do not exhibit significant limitations, as the novel nature of the approach precludes direct comparison with established benchmarks

**Methods And Evaluation Criteria:**

The authors' effort in constructing a large-scale dataset deserves recognition, encompassing both manual and automated methodologies. This involved systematically crawling papers related to adversarial attacks, followed by an extensive screening process. The resulting benchmark, grounded in real-world defenses, highlights the study’s commendable completeness in paper selection and methodological novelty.

**Other Comments Or Suggestions:**

See above.

**Other Strengths And Weaknesses:**

But out of scope of this paper, I think it can be improved for specifying detailed defense protocols and attack protocols for autonomously testing automated attacks~[1,2].

[1] Fu, Qi-An, et al. "{AutoDA}: Automated decision-based iterative adversarial attacks." 31st USENIX Security Symposium (USENIX Security 22). 2022.
[2] Guo, Ping, et al. "L-autoda: Large language models for automatically evolving decision-based adversarial attacks." Proceedings of the Genetic and Evolutionary Computation Conference Companion. 2024.

**Questions For Authors:**

No further questions.

**Relation To Broader Scientific Literature:**

This benchmark holds potential for extension to medical imaging domains, which could advance robustness research in the medical field.

**Theoretical Claims:**

Not applicable.

---

> ### Author Rebuttal · Authors · 2025-04-01
>
> We are glad the reviewer finds our claims substantiated, the completeness of our paper commendable and our method novel. We also thank the reviewer for providing relevant references. We will provide more details on attack and defense protocols for autonomously testing automated attacks in the camera ready.

---

### Official Review · Reviewer_vhhw · 2025-03-13

**Overall Recommendation:** 4

**Summary:**

This paper introduces AutoAdvExBench, a benchmark for evaluating large language models' ability to autonomously exploit defenses against adversarial examples. Unlike proxy security benchmarks, AutoAdvExBench directly measures LLMs' success on tasks regularly performed by machine learning security researchers. The authors curate 71 adversarial example defenses (47 real-world implementations and 24 "CTF-like" homework exercises) and design an agent to exploit them. Their key finding reveals a significant gap between benchmark types: their best agent achieves 75% success on CTF-like defenses but only 13% on real-world defenses, highligting the challenges of working with unstructured research code versus pedagogical implementations. The paper also shows that despite LLMs getting better, they can't comprehensively replace the machine learning security researcher yet.

**Claims And Evidence:**

The paper's central claim about the difficulty gap between CTF-like and real-world defenses is well-supported by the empirical results. The authors construct a compelling dataset through careful filtering of arXiv papers, manual verification, and reproduction of defense implementations. Their methodology for evaluating attack success is clear and follows standard practices in adversarial robustness research.

The claim that this benchmark provides a more realistic assessment than previous security benchmarks is reasonable, though could benefit from direct comparison with existing benchmarks like Cybench. The paper lacks evidence that success on this benchmark would translate to novel research results beyond demonstrating that current models struggle with the task.

**Essential References Not Discussed:**

No

**Experimental Designs Or Analyses:**

The experimental design is sound. The authors compare multiple LLM backends (Claude 3.5 Sonnet, GPT-4o, o1) and explore combinations (Sonnet 3.5 + o1 supervision), which provides insight into different models' capabilities. Breaking down success rates by attack stage helps diagnose where models struggle.

One limitation is the absense of a human baseline; the paper argues these defenses are challenging even for expert researchers but doesn't measure human performance on the same benchmark. This makes it difficult to contextualize the 13% success rate - is this far from human capabilities or relatively close (i.e., how would an individual human do, compared to the field overall)?

**Methods And Evaluation Criteria:**

The methods for benchmark construction are thorough and well-justified. The authors start with 612,495 arXiv papers and systematically filter down to 47 reproducible defenses. Their four-stage evaluation process (implement forward pass, make differentiable, implement FGSM, extend to PGD) follows standard adversarial attack methodologies. The evaluation metric (attack success rate measured by robust accuracy) also makes sense. Their agent design builds on established approaches for tool use and code generation, with a task-specific adaptation that improves performance. The decision to report results as CDF-like curves of defense accuracies rather than binary success/failure metrics provides more granular insight into model capabilities.

**Other Comments Or Suggestions:**

* It could be nice to see which specific defenses were successfully attacked, and to what extent they have common characteristics

**Other Strengths And Weaknesses:**

Strengths:

* The benchmark addresses an important gap in evaluating AI systems' ability to perform security-relevant tasks
* The careful curation of diverse defenses represents significant effort and contribution to the community
* Breaking down success by attack stage provides excellent diagnostic information about model capabilities

Weaknesses:

* The benchmark may have limited shelf life as models improve (this is acknowledged by the authors)
* It'd be nice to actually have a study to get the human baseline number (although this would take additional work)

**Questions For Authors:**

* Did you observe any correlation between attack success and defense complexity metrics (e.g., lines of code, number of dependencies, or code quality measures)?
* Have you considered evaluating human performance (perhaps from security experts) on a subset of these defenses to provide context for the 13% model success rate?

**Relation To Broader Scientific Literature:**

The paper positions itself at the intersection of LLM evaluation and adversarial machine learning. It builds on agentic benchmarks like SWE-Bench while addressing a specific security-relevant task. The authors discuss how their approach differs from other security benchmarks by focusing on end-to-end tasks rather than proxy metrics.

**Theoretical Claims:**

N/A

---

> ### Author Rebuttal · Authors · 2025-04-01
>
> We are happy that the reviewer believes that our benchmark addresses an important gap and recognizes the significant effort and contribution.
>
> **Would success on this benchmark translate on novel research results?** Papers that included breaks of several adversarial example defenses were published at top-tier venues \[1, 2, 3\], so we believe that, if an agent were to fully solve the benchmark, it might find results worthy of publication.
>
> **Limited shelf life**. We agree that our benchmark, like all other benchmarks, has limited shelf life. However, we believe that, once the benchmark will be solved by an agent, we will know that said agent is very likely to be able to perform tasks at the level of a machine learning security researcher.
>
> **Human baseline**. A few of the defenses were already successfully attacked by human researchers, however, they were not all evaluated in the same threat model, so a fair comparison would be hard. In general, we would expect the success rate for a human researcher to be much higher than the current 13%.
>
> **Correlation between defense complexity metrics and attack success**. We will look into this and, if we find interesting insights, will add the results to the camera-ready version.
>
> **List of successfully attacked defenses**. We have already created a webpage (which we will reference from the paper only in the camera-ready version for anonymization reasons) that contains all the traces coming from the execution of the agents on all the defenses, including how successful each model is at attacking each defense.
>
> References:
>
> - [1] Anish Athalye, Nicholas Carlini, and David Wagner. *Obfuscated gradients give a false sense of security: Circumventing defenses to adversarial examples*. ICML, 2018\.
> - \[2\] Florian Tramer, Nicholas Carlini, Wieland Brendel, and Aleksander Madry. *On adaptive attacks to adversarial example defenses*. NeurIPS 2020\.
> - \[3\] Francesco Croce, Sven Gowal, Thomas Brunner, Evan Shelhamer, Matthias Hein, and Taylan Cemgil. *Evaluating the adversarial robustness of adaptive test-time defenses*. ICML 2022

---

### Official Review · Reviewer_xwWg · 2025-03-14

**Overall Recommendation:** 4

**Summary:**

The paper proposes a new benchmark to evaluate LLMs and in particular their reasoning capabilities. The tasks purposed in the paper is an end-to-end real world task that consists in generating the code of a new attack based on and existing defence for image classification. In practice the LLM based agent has access to the paper of the defence, the implementation code, the perturbation bound and 1000 images. The evaluation metric is the success rate of the attack suggested by the agent.

## update after rebuttal

This is an interesting paper that can have significant impact on the machine learning security research. Any newly proposed defence should from now be challenged against this LLM-based attack generation approach. All reviewers are positive and I am definitely supporting acceptance of this work.

**Claims And Evidence:**

The paper claims that benchmarks should consider real-world end-to-end tasks and not predefined exercises-like (i.e. CTF like) tasks, as the latest is not a good proxy for the former. This claim is verified in the experiments: existing approach are effective on exercises-like tasks, (75%) but the success rate vanishes for real world tasks (13%) while the goal of the task remain the same (generating the code for adversarial attack again a specific defence).

The claim that the benchmark is challenging and can support future research is supported by the fact that current generic approaches and specialised approach (presented in 5.1) are still failing on the task of breaking defences (13% success rate).

The claim that the benchmark is tractable is demonstrated in section 5.2. The paper proposes metrics for each of the four identified steps to solve the problem of the benchmark, and reports the quantitative results for each step.

**Essential References Not Discussed:**

None that I know of.

**Experimental Designs Or Analyses:**

The experiments are well-designed.

**Methods And Evaluation Criteria:**

The evaluation criteria make sense and are inline with real-world expectations.

My only concern regards the capacity of the fact that the evaluation metric (attack success rate) my not reflect the capacity of the LLM agent to solve the exact task but a similar task (e.g. developing a better attack) that has the same objective (augmenting the success rate of the attack). See weakness.

**Other Comments Or Suggestions:**

The paper or the appendix could benefit from an end-to-end example of the defence at hand and the results of the LLM agent.

**Other Strengths And Weaknesses:**

Strengths:
- The paper is well written
- The paper tackles the interesting problem of benchmarking LLM based agents on real-world multistep tasks. The method proposed in sound and could help the development of LLMs capable of reasoning on complex tasks.
- I appreciate the real world task approach to develop benchmark.

Weaknesses:
- The evaluation metrics may be de-correlated from the defence mechanism. We have no guarantee on the method used by the attack to obtain a drop of the robust accuracy of the target model. How to guarantee this is the LLM by-passing the method in contrast to the LLM developing a stronger attack independently of the defence (i.e. regardless of the prompt).

**Questions For Authors:**

No question

**Relation To Broader Scientific Literature:**

The paper is well grounded in the two side of the literature it uses and contributes: image adversarial attacks and defenses and LLM benchmarks.

**Theoretical Claims:**

N/A

---

> ### Author Rebuttal · Authors · 2025-04-01
>
> We are glad that the reviewer appreciates the real-world oriented approach of our benchmark. We address the questions as follows:
>
> **Evaluation metric**. Can the reviewer please clarify what they mean more precisely? Does the reviewer believe that the steps that we ask the models to do (i.e., implement a differentiable forward pass, then FGSM and then PGD) might be too restrictive, or that the model might find a way to hack the framework and increase the ASR without really breaking the defense?
>
> We believe that the guiding steps are broad enough for the model to implement any strategy that may be required to solve the problem. Additionally, we enforce an epsilon bound on the perturbations that the model can produce (see “Output” paragraph in section 4). This makes sure that the model cannot hack the evaluation by e.g. turning everything into random noise.
>
> **End-to-end example**. We have already created a webpage (which we will reference from the paper only in the camera-ready version for anonymization reasons) that contains all the traces coming from the execution of the agents on all the defenses.

---

> > ### Comment · Reviewer_xwWg · 2025-04-07
> >
> > Dear authors,
> >
> > Thank you for your rebuttal and clarifying your plan to release a webpage with all execution traces.
> >
> > My point regarding the metrics:  there are currently two ways the agent can increase ASR:
> > 1. The agent "understands" the defense and breaks it with a simple specific “trick” (e.g. changing the loss/objective function). I expect this attack to significantly increase the ASR.
> > 2. The agent develops a non-specific, stronger attack that only marginally improves the ASR.
> >
> > Do you consider the two approaches equivalent to evaluate the capacity of the model to solve the problem, or is the first option the more desirable? In the latter case, how can we check it specifically breaks the given defence?

---

### Decision · Program_Chairs · 2025-05-01

**Decision:**

Accept (oral)

**Comment:**

Overall, the reviewers recognize that the paper presents a new benchmark for evaluating LLMs’ ability to autonomously bypass adversarial defenses. A rigorous assessment is conducted to evaluate LLMs’ capacity to replicate tasks typically conducted by security professionals.